# AesStyler: Aesthetic Guided Universal Style Transfer

## ABSTRACT

Recent studies have shown impressive progress in universal style transfer which can integrate arbitrary styles into content images. However, existing approaches struggle with low aesthetics and disharmonious patterns in the final results. To address this problem, we propose AesStyler, a novel Aesthetic Guided Universal Style Transfer method. Specifically, our approach introduces the aesthetic assessment model, trained on a dataset with human-assessed aesthetic scores, into the universal style transfer task to accurately capture aesthetic features that universally resonate with human aesthetic preferences. Unlike previous methods which only consider aesthetics of specific style images, we propose to build a Universal Aesthetic Codebook (UAC) to harness universal aesthetic features that encapsulate the global aspects of aesthetics. Aesthetic features are fed into a novel **U**niversal and **S**tyle-specific **Aes**thetic-Guided **A**ttention (USAesA) module to guide the style transfer process. USAesA empowers our model to integrate the aesthetic attributes of both universal and style-specific aesthetic features with style features and facilitates the fusion of these aesthetically enhanced style features with content features. Extensive experiments and user studies have demonstrated that our approach generates aesthetically more harmonious and pleasing results than the state-of-the-art methods, both aesthetic-free and aesthetic-aware.

## CCS CONCEPTS

• **Applied computing → Fine arts**; • **Computing methodologies → Rendering**; **Image manipulation**.

## KEYWORDS

Aesthetic, Universal Style Transfer, Harmonious

## 1 INTRODUCTION

Style transfer involves transferring the style of a style image $I_s$ onto a content image $I_c$ while preserving the content structure of $I_c$ simultaneously. The seminal work of Gatys *et al.*[10–12] marked the beginning of substantial advancements in this field, with progress spanning various facets including efficiency [17, 22, 38], quality [21, 24, 30, 34, 42], generalization [2, 9, 15, 16, 23], diversity [23, 39, 41].

Universal Style Transfer (UST), a key problem in this area, strives to achieve a balance among generalization, quality, and efficiency, a triad of objectives that are often in contention [23]. Current UST methods fall into two main categories based on how they manipulate content and style features: global statistics-based (e.g.

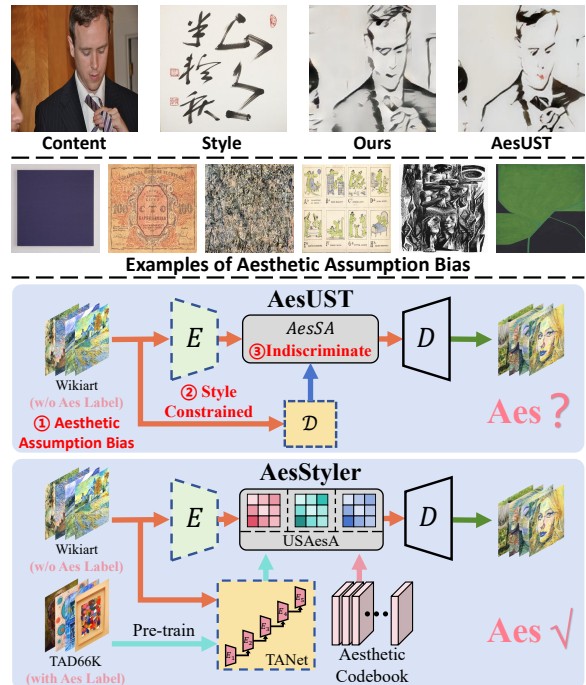

**Figure 1: Comparison of results generated by our method and AesUST, Examples of Aesthetic Assumption Bias. AesUST mainly suffers from three problems: Aesthetic Assumption Bias, Style-Constrained Aesthetic Extraction and Indiscriminate Feature Fusion.**

AdaIN [15] and ArtFlow [1]) and local patch-based (e.g. SANet [30], MAST [7], AdaAttN [27], and IECAST [3]). Although these methods achieve good results they entirely disregard aesthetic aspects of images, making them frequently generate results with aesthetically disharmonious patterns.

To mitigate this issue, AesUST [40] proposed to add a discriminator to extract aesthetic features to enhance the aesthetic aspects of style features. However, sometimes AesUST still yields results with evident aesthetic disharmonies and artifacts. This is due to the following problems: (1) *Aesthetic Assumption Bias:* In the training stage, the aesthetic discriminator of AesUST lacks explicit supervisory signals to define aesthetics, instead it presumes that images from the style training dataset inherently possess aesthetics—a presumption that is not guaranteed to be accurate as shown in Fig. 1. This can lead to it acting more as a style feature extractor, potentially overlooking true aesthetic elements. (2) *Style-Constrained Aesthetic Extraction:* AesUST restricts aesthetic feature extraction to style images, leading to a narrow, style-specific aesthetic perspective. However, aesthetics generally have universal qualities and shouldn't be confined as style-specific. (3) *Indiscriminate Feature Fusion:* attention scores recalibrate high layer feature maps of aesthetic and style features. However, these modified features are then

merged with content features without considering the differences in feature distributions and information from lower layers.

Inspired by the recent work TANet [13], which trains an aesthetic assessment model on a large-scale dataset with human-assessed aesthetic scores for precise aesthetic score prediction, we propose to leverage the strong capability of pre-trained aesthetic assessment model in accurately discerning aesthetic features to guide the style transfer process.

To this end, we propose **AesStyler**, a novel Aesthetic Guided Universal Style Transfer method. Firstly, we propose to utilize TANet as the aesthetic feature extractor in AesStyler. Secondly, we propose to build a Universal Aesthetic Codebook (UAC), to harness and utilize universal aesthetic features which encapsulate the global aspects of aesthetics. Thirdly, we propose the Universal and Style-specific Aesthetic-Guided Attention (USAesA) module. USAesA empowers our model to adaptively and progressively integrate both universal and style-specific aesthetic features with the style feature and incorporate the aes-enhanced style feature into the content feature. Extensive experiments and user studies have demonstrated the superiority of our approach. Compared to previous methods, our AesStyler not only yields results of superior aesthetics but also with better style transfer quality.

In summary, our contributions are threefold:

- We propose AesStyler, a novel Aesthetic Guided Universal Style Transfer method. We introduce the pre-trained aesthetic assessment model into UST task as the aesthetic feature extractor to accurately capture aesthetic features that resonate with human aesthetic preferences.
- We propose to build a Universal Aesthetic Codebook (UAC), harnessing universal aesthetic features that capture global aesthetic elements and later utilizing these features to guide the model to generate more universally appealing results with these features.
- We propose the Universal and Style-specific Aesthetic-Guided Attention (USAesA) module, empowering our model to adaptively and progressively integrate both universal and style-specific aesthetic features with the style feature and incorporate the aesthetics-enhanced style feature into the content feature.

## 2 RELATED WORK

### 2.1 Aesthetic-Free Neural Style Transfer

Gatys *et al.* [12] found that hierarchical layers within CNNs serve as effective tools for extracting both image content and style texture information and introduced an optimization-driven approach to iteratively generate stylized images. Certain methodologies [17, 22] have embraced an end-to-end model to facilitate real-time style transfer, tailored to a specific style. More generally, arbitrary style transfer has gained more attention in recent years. Huang *et al.* [15] introduced the concept of adaptive instance normalization (AdaIN), which adaptively applies the mean and standard deviation of each style feature to shift and re-scale the corresponding normalized content feature. Based on the CNN model, works [6, 7, 27, 30, 43] introduced self-attention into the encoder-transfer-decoder framework, enhancing the fusion of features. AdaAttN [27] proposed to take both shallow and deep features into account and properly

normalizes content feature. However, it should be noted that existing style transfer methods often overlook the aesthetic aspects of their outputs. In contrast, our proposed model strikes a harmonious balance encompassing style, content, and aesthetic of the results.

### 2.2 Aesthetic-aware Neural Style Transfer

Similar to our work, Sanakoyeu *et al.* [32], Kotovenko *et al.* [19, 20], Chen *et al.* [4, 5], and Zuo *et al.* [46] have tried to use aesthetic information to guide the style transfer. However, their aesthetic assessment models are developed to discriminate the aesthetics associated with specific artists, for instance, Claude Monet, whereas our model encapsulates artist-independent and universal aesthetics. Furthermore, their models are restricted to transferring the styles of predefined artists, whereas our approach attains universal style transfer capabilities. Furthermore, AAST [14] proposed aesthetic-aware style transfer, albeit with aesthetics defined as a fusion of color and texture alone. AesUST [40] tries to integrate aesthetic information into the style transfer process. However, previous works in aesthetic-guided style transfer did not use an aesthetic-annotated dataset to train aesthetic discriminators, potentially limiting their ability to learn true aesthetic patterns. In contrast, we offer a more comprehensive concept of aesthetics, utilizing an aesthetic assessment model pre-trained on the human-annotated aesthetic dataset to more effectively guide the style transfer process.

### 2.3 Image Aesthetic Assessment

Early Image Aesthetic Assessment (IAA) models mainly focused on extracting handcrafted features from images and mapping the visual features to annotated aesthetics labels either with trained binary classifiers or with regressors [8, 28]. The emergence of large-scale IAA datasets [18, 29] leads to the continued evolution of methods rooted in deep learning. NIMA [36] proposed to use the Earth Mover's Distance (EMD) loss to train the score distribution task. MPada [33] implemented an attention-based mechanism, dynamically fine-tuning the weights of individual image patches during training, thus improving the efficiency of the learning procedure. SAAN[44] proposes a style-specific artistic image assessment. In this work, we employ the state-of-the-art IAA model, TANet [13] to provide aesthetic guidance for the UST task. TANet can accurately predict aesthetic scores leveraging a target-aware network and an RGB-distribution-aware attention network.

## 3 METHOD

In this section, we will first introduce the overall framework of the AesStyler in Section 3.1 and explain the component details in Section 3.2 and loss functions in Section 3.3.

### 3.1 Overall Framework

Given a style image $I_s \in \Phi_s$ and a content image $I_c \in \Phi_c$, where $\Phi_s$ and $\Phi_c$ are the style dataset and the content dataset respectively, the goal of UST is to learn a transformation which successfully transfers the style of $I_s$ to $I_c$ while preserving the semantic information of $I_c$.

As shown in Fig. 2, our AesStyler is composed of five principal components: (1) An Encoder-Decoder Module with a pre-trained VGG-19 network [35] $E_{vgg}$ to extract multi-layer feature maps and a decoder network $D_{vgg}$ tasked with reconstructing the stylized

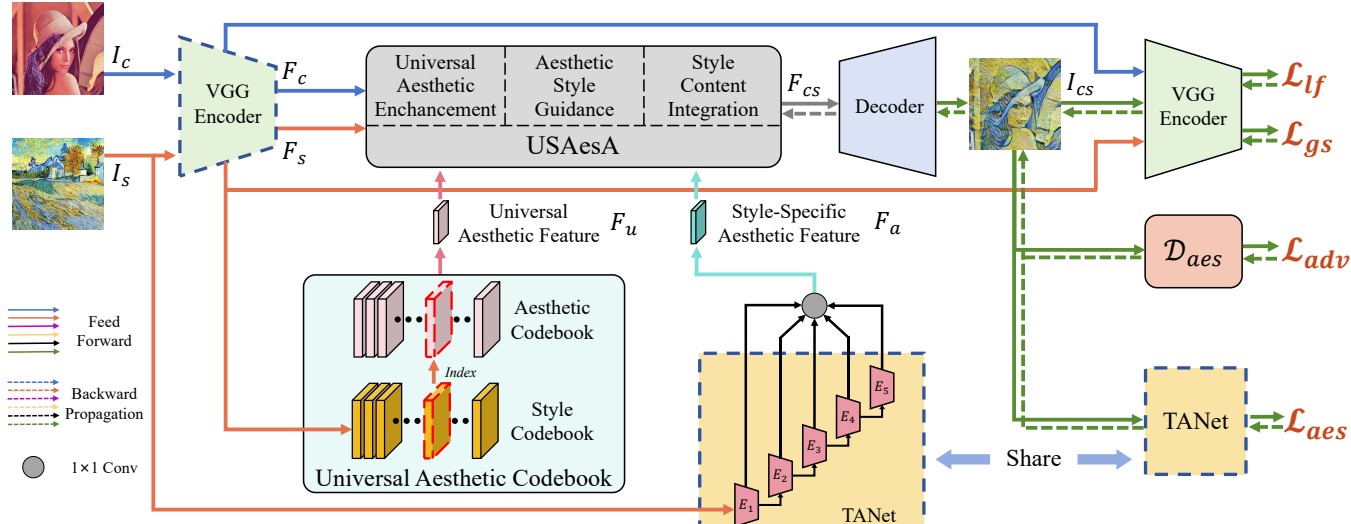

**Figure 2: Overview of our AesStyler. AesStyler primarily consists of 5 components: Encoder-Decoder Module, Aesthetic Assessment Module, Universal Aesthetic Codebook, Universal and Style-specific Aesthetic-Guided Attention Module and Aesthetic Discriminator.**

images from feature embeddings. (2) A pre-trained aesthetic assessment model, TANet [13], to precisely extract aesthetic features. (3) A novel Universal Aesthetic Codebook to offer proper universal aesthetic features. (4) A novel Universal and Style-specific Aesthetic-Guided Attention module to adaptively integrate style patterns into content features guided by universal and style-specific aesthetic features. (5) An aesthetic discriminator $\mathcal{D}_{aes}$ to circumvent the rudimentary deception of aesthetics. The pipeline is as follows:

(1) Given a pair of content image $I_c$ and style image $I_s$, we extract the VGG content features $F_c = E_{vgg}(I_c)$ and the style features $F_s = E_{vgg}(I_s)$.

(2) We then utilize the style features $F_s$ to query the proper universal aesthetic features $F_u$ from the UAC and extract the style-specific aesthetic features $F_a$ with TANet.

(3) Upon obtaining $F_c, F_s, F_g, F_a$, we input them into the USAesA module to synthesize the aesthetic-enhanced stylized features $F_{cs} = USAesA(F_c, F_s, F_u, F_a)$.

(4) Finally, the stylized result $I_{cs}$ is produce by feeding $F_{cs}$ into the decoder $D_{vgg}$, $I_{cs} = D_{vgg}(F_{cs})$.

## 3.2 Component Details

**Encoder-Decoder Module.** Similar to [15], we use the pre-trained VGG-19 network [35] as our encoder $E_{vgg}$ and freeze it during the whole training stage. The decoder $D_{vgg}$ is trainable and mirrors the encoder, albeit with the substitution of all pooling layers by nearest up-sampling layers.

**Aesthetic Assessment Module.** We employ TANet [13] to assess the aesthetic score of style transfer results and extract aesthetic features that can correctly guide the style transfer. To be specific, TANet is pre-trained with an aesthetic score regression task on TAD66K [13], a large-scale aesthetic dataset annotated with human-assessed aesthetic scores, ensuring that TANet can accurately capture aesthetic features that resonate with human perceptions of aesthetics. As to the aesthetic features, we extract

the feature maps from the *InvertedResidual-57, InvertedResidual-93, InvertedResidual-120, InvertedResidual-147, and InvertedResidual-156* layers within the *Aesthetic Perceiving Branch* of the TANet as the aesthetic features, which are then utilized in the USAesA module to guide the style transfer process. To maintain TANet's capability of detecting aesthetic patterns, we keep it frozen during the entire training process.

**Universal Aesthetic Codebook (UAC).** Besides style-specific aesthetic features, certain aesthetic features should remain universal, unaffected by variations in style. Inspired by [45], we propose to build a *Universal Aesthetic Codebook* (UAC). By utilizing images ranked top in terms of aesthetic scores in the whole style dataset and retrieving those with the closest cosine similarity, we can capture universal aesthetic features that embody the overarching elements of aesthetics while minimally disturbing the style information.

We first use TANet [13] to assess the aesthetic scores of the whole WikiArt dataset [31] and curate 1000 images with the highest aesthetic scores. Then, as demonstrated in the upper part of Fig. 3, we compute their style features via the VGG-19 network [35] for later query and extract their aesthetic features using TANet [13], which are then stored as UAC, as $\hat{F}_i$ for the $i$th selected image.

As shown in the lower part of Fig. 3, during training, we use $E_{vgg}$ to extract the style feature $F_s$ of the style image $I_s$. We use $F_s$ as the query to retrieve the corresponding universal aesthetic feature $F_u$ from the UAC. Specifically, we calculate the cosine similarity between the style feature $F_s$ and all style features $\hat{F}_i$ stored in the UAC as follows:

$$Cosine\ Similarity(F_s, \hat{F}_i) = \frac{\Gamma(F_s) \cdot \Gamma(\hat{F}_i)}{||\Gamma(F_s)|| \times ||\Gamma(\hat{F}_i)||}, \quad (1)$$

where $\Gamma$ and $\cdot$ denote the feature vectorization and dot product operations respectively, $\hat{F}_i$ refers to the $i$th style feature stored in the UAC. We compute one thousand cosine similarities, subsequently ascertaining the *index* corresponding to the image with

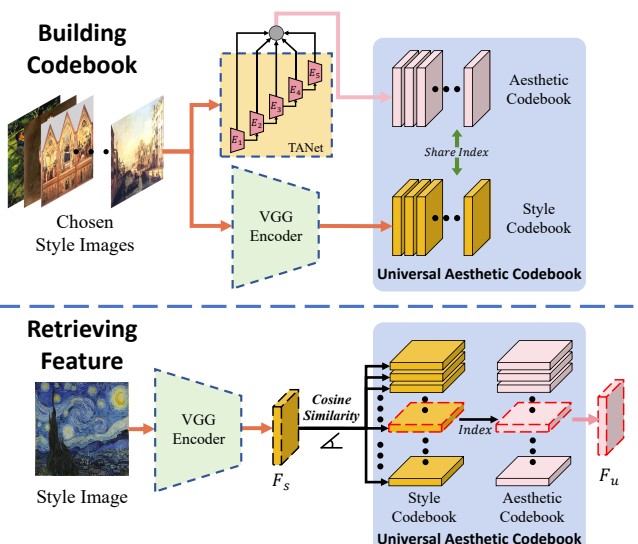

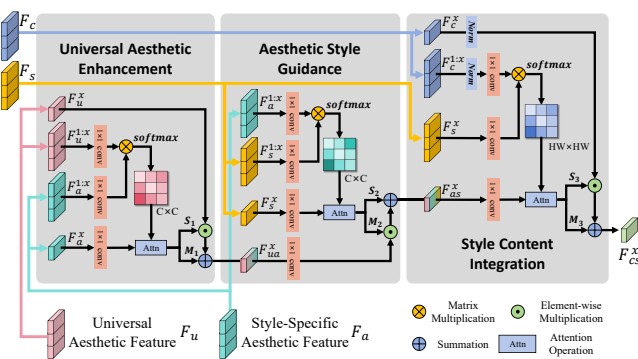

Figure 3: The processes of building the codebook and retrieving features of the Universal Aesthetic Codebook (UAC).

the maximal cosine similarity:

$$index = \arg\max_i Cosine\ Similarity(F_s, \hat{F}_i). \quad (2)$$

Thus, we can retrieve the universal aesthetic feature $F_u = \hat{F}_{index}$, which encapsulates the global aspects of aesthetics while minimally disturbing the style information. We are able to employ this feature, along with style-specific aesthetic features, to guide the model towards generating more universally appealing results.

**Universal and Style-specific Aesthetic-Guided Attention (USAesA) Module.** In order to guide the style transfer process with universal and style-specific aesthetic information, we propose a novel attention-based style transfer module, Universal and Style-specific Aesthetic-Guided Attention (USAesA) module, which can adaptively integrate style patterns into content features, considering both universal and style-specific aesthetic attributes. As shown in Fig. 4, USAesA works in three phases: (1) Given the universal aesthetic feature $F_u$ retrieved from UAC and the style-specific aesthetic feature $F_a$ extracted from the style image, we first use $F_u$ to enhance global aesthetic attributes of $F_a$. (2) We then use the universally enhanced aesthetic feature $F_{ua}$ to further integrate aesthetic information into the style feature $F_s$. (3) Lastly, we integrate the aesenhanced style feature $F_{as}$ according to the semantic distributions of the content feature $F_c$.

*Phase 1: Universal Aesthetic Enhancement.* Given the universal aesthetic feature $F_u$ retrieved from UAC and the style-specific aesthetic feature $F_a$ extracted from the style image, we first use $F_u$ to enhance global aesthetic attributes of $F_a$. Inspired by [27] that attention maps and adaptive normalization can fuse style and content information, we propose to employ these two mechanisms to globally enhance the style-specific aesthetic feature by channel distribution.

Specifically, given $F_u^{1:x}, F_u^x$, where $x$ means $x$th layer in VGG features, retrieved from the UAC and $F_a^{1:x}, F_a^x$ extracted from the style image by TANet [13], we first obtain $Q_u$ (query), $K_u$ (key) and

Figure 4: Universal and Style-specific Aesthetic-Guided Attention (USAesA) module. USAesA encompasses three distinct phases: Phase 1: Universal Aesthetic Enhancement, Phase 2: Aesthetic Style Guidance and Phase 3: Style Content Integration. $F_c, F_s, F_u$ and $F_a$ represents content, style, universal aesthetic and style-specific aesthetic features respectively.

$V_a$ (value):

$$
\begin{aligned}
Q_u &= f_u(F_u^{1:x}), \\
K_a &= f_a^1(F_a^{1:x}), \\
V_a &= f_a^2(F_a^x),
\end{aligned}
\quad (3)
$$

where $f_u, f_a^1$ and $f_a^2$ are $1 \times 1$ learnable convolutions.

Then, we calculate the aesthetic attention between $\hat{F}_u^{1:x}$ and $\hat{F}_a^{1:x}$ according to the following equation:

$$A_1 = Softmax(Q_u \otimes (K_a)^T), \quad (4)$$

where $\otimes$ denotes matrix multiplication and $T$ denotes the matrix transpose operation. We then calculate the attention-weighted mean and standard deviation respectively:

$$
\begin{aligned}
M_1 &= A_1 \otimes V_a, \\
S_1 &= \sqrt{A_1 \otimes (V_a \circ V_a) - M_1 \circ M_1},
\end{aligned}
\quad (5)
$$

where $\circ$ denotes the element-wise product.

Finally, for each position and channel of the universal aesthetic feature, the corresponding scale in $S_1$ and shift in $M_1$ are used to generate the final universally enhanced aesthetic feature:

$$F_{ua}^x = S_1 \circ F_u^x + M_1. \quad (6)$$

Thus, we effectively enhance the style-specific aesthetic feature by endowing it with more universal aesthetic qualities which resonate with human aesthetic preferences

*Phase 2: Aesthetic Style Guidance.* After using the universal aesthetic feature to enhance the style-specific aesthetic feature, we integrate the universally enhanced aesthetic information into the style features. Similarly, given aesthetic features $F_a^{1:x}, F_{ga}^x$ and style features $F_s^{1:x}, F_s^x$ extracted by the VGG-19 network [35], we first obtain $Q_a, K_s$ and $V_s$, the attention map between $Q_a$ and $K_s$, the attention-weighted mean and standard deviation respectively:

$$
\begin{aligned}
Q_a &= f_a^3(F_a^{1:x}), K_s = f_s^1(F_s^{1:x}), V_s = f_s^2(F_s^x), \\
A_2 &= Softmax(Q_a \otimes (K_s)^T), \\
M_2 &= A_2 \otimes V_s, \\
S_2 &= \sqrt{A_2 \otimes (V_s \circ V_s) - M_2 \circ M_2},
\end{aligned}
\quad (7)
$$

where $f_a^3, f_s^1$ and $f_s^2$ are $1 \times 1$ learnable convolutions.

Finally, we scale and shift the universally enhanced aesthetic feature with $S_2$ and $M_2$ respectively to generate the final aesthetically enhanced style feature:

$$F_{as}^x = S_2 \circ f_{ua}(F_{ua}^x) + M_2, \tag{8}$$

where $f_{ua}$ is $1 \times 1$ learnable convolution.

*Phase 3: Style Content Integration.* Upon refining the style feature with aesthetic guidance, our objective is to integrate them into the content feature, thereby accomplishing an aesthetic-aware style transfer. We transfer feature statistics via generating attention-weighted mean and standard variance maps. Given aesthetic-enhanced style features $F_{as}^x$, lower-layer style features $F_s^{1:x}$ and content features $F_c^{1:x}, F_c^x$, we first obtain $Q_c$, $K_s'$ and $V_{as}$, the attention map between $Q_c$ and $K_s'$, the attention-weighted mean and standard deviation respectively:

$$Q_c = f_c(F_c^{1:x}), K_s' = f_s^3(F_s^{1:x}), V_{as} = f_{as}(F_{as}^x),$$
$$A_3 = Softmax(Q_c \otimes (K_s')^T),$$
$$M_3 = A_3 \otimes V_{as},$$
$$S_3 = \sqrt{A_3 \otimes (V_{as} \circ V_{as}) - M_3 \circ M_3}, \tag{9}$$

where $f_c, f_s^3$ and $f_{as}$ are $1 \times 1$ learnable convolutions.

Finally, we scale and shift the normalized content feature with $S_3$ and $M_3$ respectively to generate the final result:

$$F_{cs}^x = S_3 \circ Norm(F_c^x) + M_3. \tag{10}$$

USAesA enables our model to adaptively and simultaneously integrate the aesthetic attributes of both universal and style-specific aesthetic features with the style feature according to the global aesthetic channel distribution, and subsequently aids in the integration of the aesthetically enhanced style feature into the content feature, in harmony with the semantic spatial distribution of the content feature.

**Aesthetic Discriminator.** As previous works [26, 37] have shown that aesthetic attributes of one image are closely related to its textures, we find that if we directly optimize the aesthetic scores of images, it may lead to strange textures, which we conjecture act as a primitive guise intended to deceive the TANet into judging the results as aesthetically pleasing. To solve this problem, we propose to employ a new aesthetic discriminator $\mathcal{D}_{aes}$ which plays the min-max game of discriminating between real artworks from WikiArt dataset [31] and style transfer results along with the generator to avoid the appearance of strange textures. It is important to highlight that the aesthetic discriminator we introduce differs from the one presented in AesUST [40]. Our Aesthetic Discriminator primarily targets and mitigates odd textures, which could mislead TANet into deeming the results aesthetically pleasing. In contrast, the aesthetic discriminator in AesUST aims to preserve aesthetics, yet it encounters an aesthetic assumption bias, as discussed in Section 1.

## 3.3 Loss Function

*Aesthetic loss.* To augment the aesthetic quality of the style transfer results, we employ the negative aesthetic score as the aesthetic loss term:

$$\mathcal{L}_{aes} = -TANet_{score}(I_{cs}), \tag{11}$$

where *TANet* refers to the pre-trained aesthetic assessment model TANet [13] which will produce an aesthetic score for the style transfer result $I_{cs}$.

*Adversarial Loss.* In order to make the style transfer results look similar to real artistic images and avoid artifacts related to textures, the newly proposed discriminator $\mathcal{D}_{aes}$ plays the min-max game with the generator as follows:

$$\max_{\mathcal{D}_{aes}} \min_{\mathcal{G}} \mathcal{L}_{adv} = \mathbb{E}_{I_s \sim \Phi_s}[\log(\mathcal{D}_{aes}(I_s))]$$
$$+ \mathbb{E}_{I_c \sim \Phi_c, I_s \sim \Phi_s}[log(1 - \mathcal{D}_{aes}(\mathcal{G}(I_c, I_s)))], \tag{12}$$
$$\mathcal{G}(I_c, I_s) = D_{vgg}(AesStyler(E_{vgg}(I_c), E_{vgg}(I_s),$$
$$TANet(I_s), UAC(E_{vgg}(I_s)))).$$

Here the generator $\mathcal{G}$ consists of $E_{vgg}, D_{vgg}$ and *USAesA*.

*Style Loss.* Following [27], distances of mean $\mu$ and standard deviation $\sigma$ between generated images and style images in VGG feature space are penalized:

$$\mathcal{L}_s = \sum_{x=2}^{5}(||\mu(E_{vgg}^x(I_{cs})) - \mu(F_s^x)||_2$$
$$+ ||\sigma(E_{vgg}^x(I_{cs})) - \sigma(F_s^x)||_2) \tag{13}$$

*Content Loss.* $\mathcal{L}_c$ constrains the consistency between features of stylized images and transformation results:

$$\mathcal{L}_c = \sum_{x=3}^{5}\left\| E_{vgg}^x(I_{cs}) - USAesA^*(F_c, F_s) \right\|_2, \tag{14}$$

where $USAesA^*$ serves as a supervision signal that should be deterministic. Thus, we consider the parameter-free version of $USAesA^*$ without learnable $1 \times 1$ convolution layers and the aesthetic integration branch.

To conclude, our final objective is:

$$\min_{\mathcal{G}} \mathcal{L}_{\mathcal{G}} = \lambda_s \mathcal{L}_s + \lambda_c \mathcal{L}_c + \lambda_{aes} \mathcal{L}_{aes} + \lambda_{adv} \mathcal{L}_{adv},$$
$$\max_{\mathcal{D}_{aes}} \mathcal{L}_{\mathcal{D}} = \lambda_{adv} \mathcal{L}_{adv}. \tag{15}$$

# 4 EXPERIMENTS

## 4.1 Implementation Details

**Dataset.** The Aesthetic Discriminator is trained on TAD66K [13]. TAD66K is a large-scale aesthetic dataset that contains 14900 artistic images and 45100 other images annotated with human-assessed aesthetic scores. This pre-training ensures that TANet can accurately capture aesthetic features that resonate with human perceptions of aesthetics. The main network is trained with 82,783 images from MS-COCO [25] as the content dataset $\Phi_c$ and 79,433 images from WikiArt [31] as the style dataset $\Phi_s$.

**Hyper-parameter.** We set the hyperparameters $\lambda_s, \lambda_c, \lambda_{aes}$, and $\lambda_{adv}$ to 0.4, 3.5, 1.2, and 5.0, respectively, which are the optimal values obtained from experiments for the best performance.

**Details.** We employ the Adam optimizer with an initial learning rate of 0.0001, and each batch comprises 6 content and 6 style images. The model is trained for 25,000 iterations. During the training stage, all the images are uniformly resized to $512 \times 512$ and then randomly cropped to $256 \times 256$. All experiments are carried out with an NVIDIA GeForce RTX 3090 24GB GPU.

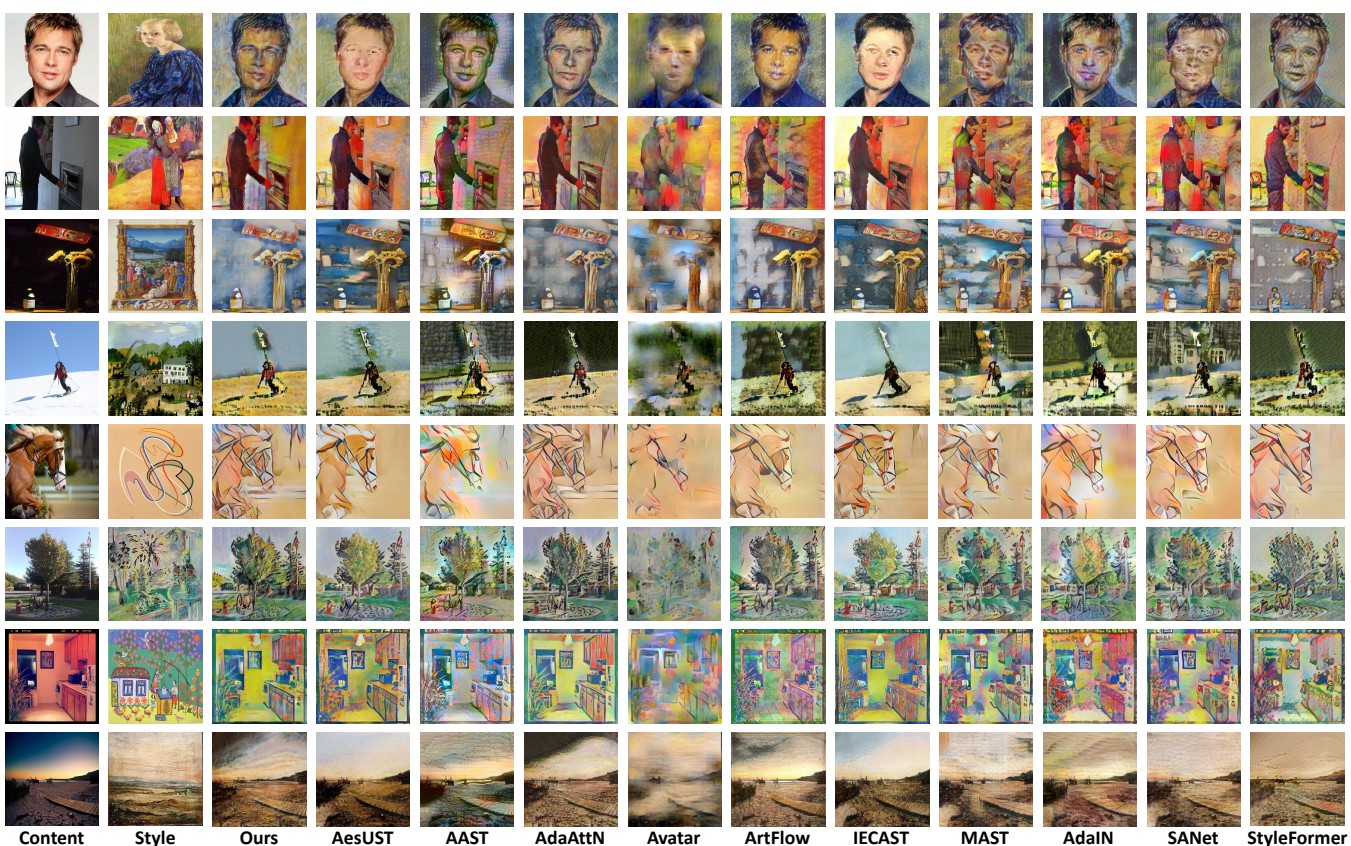

**Figure 5: Qualitative comparisons with previous state-of-the-art UST methods.**

**Table 1: Quantitative comparisons with previous state-of-the-art UST methods. Bold and Underline indicate best and second-best results. Our method achieves best Gram Loss, Aes Score and Deception Rate and second-best SSIM. Despite AdaAttN achieving the highest SSIM, its performance in Gram Loss is suboptimal, indicating that our method strikes a better balance between content, style and aesthetic.**

|  | Ours | AesUST | AAST | AdaAttN | Avatar | ArtFlow | IECAST | MAST | AdaIN | StyleFormer |
|---|---|---|---|---|---|---|---|---|---|---|
| Gram Loss ↓ | **0.1710** | 0.2192 | 0.1756 | 0.2088 | 0.2614 | 0.2046 | 0.2641 | 0.1916 | 0.1913 | 0.1713 |
| SSIM ↑ | 0.3971 | 0.3330 | 0.2780 | **0.4311** | 0.2449 | 0.3966 | 0.3392 | 0.2945 | 0.2668 | 0.3354 |
| Aes Score ↑ | **0.4597** | 0.4102 | 0.4020 | 0.4180 | 0.4100 | 0.4056 | 0.4137 | 0.4065 | 0.4046 | 0.4109 |
| Deception Rate ↑ | **0.2857** | 0.1885 | 0.2176 | 0.2761 | 0.2620 | 0.1846 | 0.1811 | 0.2730 | 0.1363 | 0.2330 |

## 4.2 Comparisons

We compare our proposed AesSTyler against 10 state-of-the-art arbitrary style transfer methods: aesthetic-aware UST methods (AesUST [40] and AAST [14]), aesthetic-free UST methods (AdaAttN [27], Avatar [34], ArtFlow [1], IECAST [3], MAST [7], AdaIN [15], SANet [30] and StyleFormer [43]).

**Qualitative Comparison**

*Style Transfer Comparison.* We first provide the qualitative comparison results in Fig. 5. Due to the rudimentary alignment of mean and variance, AdaIN [15] generates results with crack-like artifacts (2nd, 4th and 5th rows). Avatar [34] produces results with a blurred appearance and noticeable patchiness due to patch matching strategies, evident in all rows. SANet [30] and MAST [7] meticulously

transfer style features onto content features only within the deeper layers, leading to compromised content structures and muddled textures (1st, 2nd, 4th, 5th, 8th rows). The constrained feature representation capacity of flow-based models means that the outcomes of ArtFlow [1] typically suffer from a lack of style richness or accuracy (4th and 5th rows). Additionally, the borders of stylized images may exhibit undesirable patterns stemming from numerical overflow (4th, 5th and 7th rows). As to AdaAttN [27] and IECAST [3], some results show the style degradation problem, rendering the stylized patterns in the generated images inconsistent with those of the input reference in the 1st, 2nd, 4th and 5th rows and 1st, 3rd, 5th and 7th rows respectively. The outputs of StyleFormer [43]

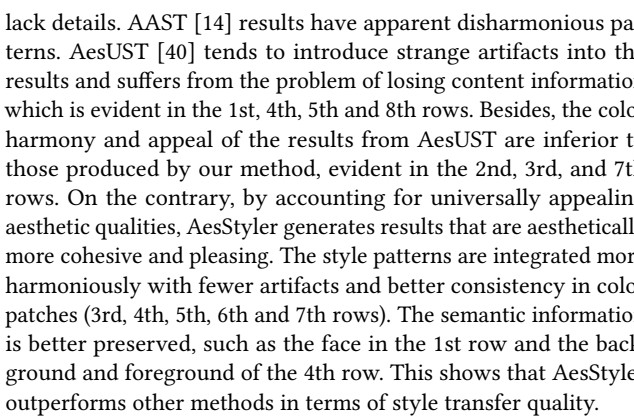

**Figure 6: Qualitative comparisons of aesthetics with previous state-of-the-art UST methods.**

lack details. AAST [14] results have apparent disharmonious patterns. AesUST [40] tends to introduce strange artifacts into the results and suffers from the problem of losing content information which is evident in the 1st, 4th, 5th and 8th rows. Besides, the color harmony and appeal of the results from AesUST are inferior to those produced by our method, evident in the 2nd, 3rd, and 7th rows. On the contrary, by accounting for universally appealing aesthetic qualities, AesStyler generates results that are aesthetically more cohesive and pleasing. The style patterns are integrated more harmoniously with fewer artifacts and better consistency in color patches (3rd, 4th, 5th, 6th and 7th rows). The semantic information is better preserved, such as the face in the 1st row and the background and foreground of the 4th row. This shows that AesStyler outperforms other methods in terms of style transfer quality.

*Aesthetic Comparison.* We also show the results of aesthetic comparison in Fig. 6. Another Universal Style Transfer method that considers aesthetics, AesUST, evidently falls short in generating images with sufficient aesthetic appeal. Artifacts in the 2nd, 3rd, and 4th rows contribute to a decline in aesthetic scores. Furthermore, the overly dark tone in the 2nd and 4th rows also leads to a reduction in aesthetic appeal. On the contrary, it is evident that our AesStyler generates results with markedly superior aesthetic quality. On the one hand, AesStyler avoids the introduction of conspicuous artifacts and overly complex patterns in the final results (1st and 3rd rows). On the other hand, AesStyler ensures color consistency and aesthetic harmony in the final results (2nd, 3rd, 4th and 5th rows). Finally, our method distinctly produces images with more vibrant colors and greater contrast (2nd and 5th rows), which are typically more pleasing to human eyes.

**Quantitative Comparison**

*Quantitative Metrics.* We present quantitative results in Table 1. *Gram loss*, which computes the difference in mean and standard deviation between Gram matrices of style and stylized features, indicating the degree of style preservation. *Structural Similarity Index Measure* (SSIM) considers changes in structural information,

**Table 2: Results of user studies. Note that 3.0 is the full score. Our method outperforms previous state-of-the-art methods, achieving the highest Style Transfer Score and Aesthetic Score.**

|  | Style Transfer Score ↑ | Aesthetic Score ↑ |
|---|---|---|
| Ours | **2.7419** | **1.5725** |
| AesUST | 0.5591 | 1.4370 |
| AdaAttN | 1.2215 | 1.1354 |
| MAST | 0.3010 | 0.1112 |
| IECAST | 0.8666 | 1.3306 |
| StyleFormer | 0.3096 | 0.4129 |

luminance, and contrast, representing the extent of content preservation. *Aesthetic score* assessed by TANet [13] indicates the aesthetic level of the final results. Gram loss, SSIM and aesthetic score are all calculated with results produced by 1000 style and content image pairs. As to *Deception Rate*, following [32], we train a VGG-16 network [35] on the WikiArt [31] dataset to classify the artist labels. The Deception Rate is calculated as the ratio of generated images that the network misclassifies as the original artworks of the artist who created the style image, which is computed using 18 style images, each emblematic of a distinct artist, paired with 300 content images, culminating in a total of 5,400 style transfer results for evaluation.

*Gram Loss & SSIM.* As delineated in Table 1, AesStyler has secured the best Gram loss and the second-best SSIM, indicating that AesStyler adeptly strikes a balance between style transformation and content preservation. Although AdaAttN achieved the highest SSIM score, its Gram loss performance is poor. As discussed in Section 4.2, AdaAttN tends to over-preserve content information, including elements related to color and style from the content images, which contributes to the style degradation problem. Furthermore, it is logical for the SSIM metric of AesStyler to be slightly lower than that of AdaAttN, as achieving aesthetic excellence necessitates the compromise of certain content fidelity.

*Aesthetic Score.* As shown in Table 1, AesStyler achieved the highest aesthetic score, indicating that our method produces results that align more closely with human aesthetic preferences. This is attributed to the strategic incorporation of aesthetic features into the style transfer process and the efficacy of the feature integration module. However, AesUST, another Universal Style Transfer method that takes aesthetics into account, clearly underperforms in aesthetic scoring. This shortfall is attributed to the Aesthetic Assumption Bias and the Style-Constrained Aesthetic Extraction, which we analyze and discuss in Section 1. Moreover, the quantitative results are also consistent with those shown in the qualitative comparison in Fig. 6, which indicates that AesStyler, along with AdaAttN, renders results that are aesthetically superior to those of other style transfer methods.

*Deception Rate.* AesStyler achieved best in the deception rate, signifying that through the assistance of both universal aesthetic features and style-specific aesthetic features, coupled with a novel feature integration mechanism, our model demonstrates a stronger capacity to produce results that bear a closer resemblance to authentic artist-created paintings.

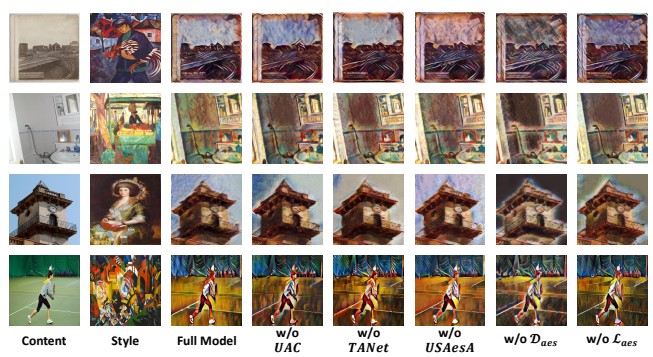

**Figure 7: Qualitative results of ablation studies. w/o *UAC*, w/o *TANet* and w/o *USAesA* indicate removing $F_u$, $F_a$ and $F_u$&$F_a$.**

## 4.3 User study

Besides qualitative and quantitative comparison, we conduct 2 user studies to further compare our method with baselines regarding the quality of style transfer and the aesthetic appeal of style transfer results. We choose AesUST [40], AdaAttN [27], MAST [7], IECAST [3] and StyleFormer [43] as baseline methods in user studies due to their commendable qualitative and quantitative performance.

In the first user study aiming to compare the style transfer quality, we invited 50 participants to respond to 15 questions. Each question presented a pair of content and style images alongside six images produced by AesStyler and the five aforementioned methods. Users were asked to rank the top-3 images that have the best style transfer quality, striking a good balance between style transformation and content preservation. The 1st, 2nd and 3rd images are given scores of 3, 2 and 1 respectively and others are scored 0. We calculate average scores for 6 methods and report them in Table 2. From these results, we can conclude that AesStyler, with the support of aesthetic features and a novel feature integration mechanism, achieves almost full score (3), signifying its capacity to realize superior style transfer quality.

In the second user study aiming to compare the aesthetic appeal of style transfer results, we invited 50 participants to answer 20 similar ranking questions. This time, however, participants were presented solely with style transfer results produced by six methods, as the intention is for the users to exclusively evaluate aesthetic appeals. The aesthetic scores for six methods are presented in Table 2. From Table 2, we can conclude that AesStyler produces results that are aesthetically more harmonious and pleasing.

## 4.4 Ablation Study

We conduct ablation studies on the UAC, TANet, USAesA, aesthetic discriminator and aesthetic loss. We show the qualitative results in Fig. 7 and report the quantitative results in Table 3. All experimental settings follow as before.

*UAC, TANet and USAesA.* Without UAC, TANet, USAesA means removing $F_u$, $F_a$ and $F_u$&$F_a$ in style transfer process respectively. Without UAC, the results show some aesthetically unpleasant features originating from the given style image (3rd and 4th rows), which can be ascribed to the fact that, without UAC, AesStyler

**Table 3: Quantitative results of ablation studies. w/o *UAC*, w/o *TANet* and w/o *USAesA* denotes removing $F_u$, $F_a$ and $F_u$&$F_a$ respectively. Deception stands for Deception Rate.**

| | Full Model | w/o *UAC* | w/o *TANet* | w/o *USAesA* | w/o $\mathcal{D}_{aes}$ | w/o $\mathcal{L}_{aes}$ |
|---|---|---|---|---|---|---|
| Gram ↓ | **0.1710** | 0.1885 | 0.1932 | 0.1841 | 0.4418 | 0.2161 |
| SSIM ↑ | **0.3971** | 0.3925 | 0.3924 | 0.3891 | 0.2989 | 0.3959 |
| Aes-Score ↑ | 0.4597 | 0.4401 | 0.4361 | 0.4353 | **0.7846** | 0.4003 |
| Deception ↑ | **0.2857** | 0.2783 | 0.2781 | 0.2746 | 0.0433 | 0.2733 |
| User-Score ↑ | **1.9730** | 1.0133 | 0.8133 | 0.7066 | 0.5600 | 0.9333 |

solely considers style-specific aesthetic features, thus bringing aesthetic deficiency in style images into final results. Without TANet, the results apparently lack aesthetic appeal (2nd and 4th rows). The absence of USAesA leads to the style degradation problem (2nd and 4th rows). Furthermore, the omission of UAC, TANet, USAesA does not improve the Gram loss and SSIM, which, by another measure, corroborates that the incorporation of these modules in guiding the style transfer process confers only benefits upon the results.

*Aesthetic Discriminator.* Without the discriminator module, although the aesthetic scores of the results increase markedly in Table 3, in Fig. 7, the images display distinctly odd textures (especially evident upon zooming in on all rows), which we surmise serve as a rudimentary subterfuge to mislead the TANet into deeming the results aesthetically pleasing. Furthermore, as shown in Table 3, the user study also reveal that, in the absence of $\mathcal{D}_{aes}$, the results demonstrate a reduced aesthetic appeal.

*Aesthetic Loss.* Without the aesthetic loss term, the results generated by our method apparently show lower aesthetic appeal than results of the full model with weird big dark patches, apparent in the 3rd, 4th and 5th rows. Besides, removing the aesthetic loss term also causes the style degradation problem, resulting in failing to capture the true style of style images.

## 5 CONCLUSION

In this paper, we propose AesStyler, a novel Aesthetic Guided Universal Style Transfer method. Our AesStyler, by utilizing TANet [13] as the aesthetic feature extractor, can accurately capture aesthetic features that resonate with human aesthetic preferences. Secondly, we propose to build a Universal Aesthetic Codebook (UAC) to harness universal aesthetic features that encapsulate the global aspects of aesthetics and to employ these features to guide the style transfer process. Thirdly, we propose Universal and Style-specific Aesthetic-Guided Attention (USAesA) module, which empowers our model to adaptively and progressively integrate both universal and style-specific aesthetic features with the style feature and incorporate the aes-enhanced style feature into the content feature. Extensive experiments and user studies have demonstrated the superiority of our method. Compared to state-of-the-art methods of both aesthetic-free and aesthetic-aware, AesStyler yields results of superior aesthetics and better style transfer quality.

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
