# OpenReview forum: "AesStyler: Aesthetic Guided Universal Style Transfer"
_acmmm.org/ACMMM/2024/Conference — MM2024 Poster_

### Official Review · Reviewer_WWaM · 2024-05-13

**Rating:** 4
**Confidence:** 3

**Summary:**

The authors presented a new aesthetic guided style transfer method, called AesStyler.

(1) Utilizing TANet as the aesthetic feature extractor, AesStyler effectively captures aesthetic features that align with human aesthetic preferences.

(2) They develop a Universal Aesthetic Codebook (UAC), which aggregates universal aesthetic features representing the global aspects of aesthetics. These features are then used to direct the style transfer process.

(3) They introduce the USAesA module. USAesA allows AesStyler to adaptively and progressively blend both universal and style-specific aesthetic features with the style feature, integrating this aes-enhanced style feature into the content feature for superior style transfer results.

**Strengths:**

The authors designed several aesthetic-aware modules to generate aesthetically more harmonious and pleasing results. The writing of this paper is logically coherent and the expression is easy to understand.

**Limitations:**

The most recent neural style transfer (NST) work cited in the related work section is from 2022, which is no longer considered state-of-the-art. It is advisable to compare more current NST methods, such as GAN-based and diffusion techniques, in terms of quality.

The primary focus of this paper is $\mathbf{aesthetic}$-guided universal style transfer (UST). To demonstrate the effectiveness of AesStyler, it is essential to compare it with more aesthetic-aware methods. Currently, only two such methods are evaluated, which may not sufficiently establish its advantages.

I suggest that the user study methods should be aesthetic-aware approaches, as these would be more convincing. Additionally, could you clarify the background of the participants? Those with a background in aesthetics would be more suitable. What exactly is the definition of aesthetic features? How participants evaluate aesthetic appeals.

It seems there is no explanation for the pictures in the second row of Figure 1in the main text.

In Phase 2: Aesthetic Style Guidance.  $F_{ga}^x$ is $F_a^x$?

**Suitability:**

2

---

### Official Review · Reviewer_1qV1 · 2024-05-15

**Rating:** 3
**Confidence:** 4

**Summary:**

The manuscript introduces "AesStyler," a novel approach for aesthetic-guided universal style transfer. Building on the existing TANet model, AesStyler enhances style transfer outputs by integrating aesthetics assessed by humans. This is achieved through the Universal and Style-specific Aesthetic-Guided Attention (USAesA) module, which merges universal and style-specific aesthetic features with traditional style features.

**Strengths:**

The integration of a pre-trained aesthetic assessment model is a clever solution to the common issue of aesthetic assumption bias in style transfer. This allows for a more nuanced capture of aesthetic features that align with broad human preferences.

**Limitations:**

1-	The reliance on TANet for extracting aesthetic features could restrict the model’s adaptability to new datasets or evolving definitions of aesthetics.

2- TANet is just used as an aesthetic feature extractor, try something new, for example Q-Align[1].
    [1] Wu et al. Q-Align: Teaching LMMs for Visual Scoring via Discrete Text-Defined Levels, The Forty-first International Conference on
        Machine Learning(ICML), 2024.

3-	The use of VGG as the backbone architecture raises questions. Exploring or justifying the choice over potentially more suitable architectures like ResNet, ResNext, ViT and Mamba could strengthen the manuscript.

4-	The core methodology primarily extends TANet without substantial modifications tailored to style transfer, potentially weakening the novelty of the approach.

5-	The manuscript details an experimental procedure involving resizing images to 512x512 pixels and randomly cropping them to 256x256. This raises concerns about whether such transformations could alter the aesthetic characteristics of the images. The review does not address how the method ensures the aesthetic properties remain consistent post-transformation, which is crucial for validating the effectiveness of the aesthetic integration. This oversight might impact the perceived reliability of the results, and further clarification on this aspect would strengthen the manuscript.

6-	While AesStyler excels in enhancing aesthetic appeal, it does not perform as well on the Structural Similarity Index Measure (SSIM) compared to AdaAttN, indicating room for improvement in maintaining structural integrity during style transfers.

**Suitability:**

3

---

### Official Review · Reviewer_5GcG · 2024-05-20

**Rating:** 4
**Confidence:** 4

**Summary:**

In this paper, a general image style transfer method based on aesthetic guidance is proposed to improve the beauty and harmony of the result of style transfer. To this end, the author introduces an aesthetic evaluation model in the process of style transfer, which captures the characteristics of human aesthetic preference resonance by establishing a universal aesthetic codebook. In addition, an attention module is constructed to guide the integration of general aesthetic characteristics and specific stylistic aesthetic characteristics. Experiments and user studies have shown that the method produces state-of-the-art results aesthetically.

**Strengths:**

1.	The paper is easy to follow;
2.	The performance of the proposed method seems great;
3.	The strengths of this paper lie in the originality of its ideas and the network design developed based on these concepts.

**Limitations:**

The author's reference to AesUST's work in section 1, where there is a problem with aesthetic assumption bias, refers to the fact that the images in the style training dataset are inherently aesthetic. In the work of this paper, the design of this network is also designed with this assumption in mind, and I am not sure why UAC is used.
	In the first phase of the USAesA module mentioned in Section 3.2, the author uses general aesthetic features to enhance specific aesthetic features, does the author think that general aesthetic features can better express aesthetics, what is the difference between general aesthetic features and specific aesthetic features, and what is the significance of this? It is recommended to emphasize this point.
	Do the content loss and style loss described in Section 3.3 correspond to L_lf and L_gs in Figure 2 ? If yes, it is better to keep them consistent; if not, please explain L_(lf ) and L_gs.
	Since the aesthetic evaluation model introduced in this work is designed based on TANet, it is better to use different aesthetic evaluation methods in the aesthetic score evaluation of quantitative experiments, which is more persuasive.
	It is recommended to evaluate the effectiveness of each component of USAesA in ablation experiments.

**Suitability:**

3

---

### Official Review · Reviewer_a814 · 2024-05-24

**Rating:** 4
**Confidence:** 4

**Summary:**

This paper presents a new Aesthetic Guided Universal Style Transfer method, which includes UAC and USAesA modules to accurately
capture aesthetic features that universally resonate with human aesthetic preferences. Extensive experiments and user studies have demonstrated the superiority of the proposed method.

**Strengths:**

- The idea of this paper is interesting.
- The structure of this paper is well organized.

**Limitations:**

- Recently, diffusion models have achieved very good results in image generation and editing, and the authors need to explain the superiority of the proposed framework compared to diffusion models.
- The authors used the pre-trained aesthetic assessment model in the UST task as the aesthetic feature extractor. However, different methods may have different results. The authors need to explain why TANet is adopted instead of other methods such as NIMA [1], TAVAR [2] and VILA [3].

  [1] NIMA: Neural Image Assessment, TIP.

  [2] Theme-aware Visual Attribute Reasoning for Image Aesthetics Assessment, TCSVT.

  [3] VILA: Learning Image Aesthetics From User Comments With Vision-Language Pretraining, CVPR.

- The proposed method seems similar to AesUST, and the authors need to explain the differences between the two, including motivation and methodology.
- There is no standard format for references; for example, authors' names and conference abbreviations need to remain consistent.
- The experimental results show that the proposed model has a good effect. To improve the contribution, it is recommended that the author open-source the code and related weights.

**Suitability:**

3

---

### Meta-Review · Area_Chair_NGQh · 2024-07-02

**Recommendation:** Accept (Poster)
**Confidence:** 5

**Metareview:**

The paper received three Borderline Accept ratings. While the reviewers acknowledged the novelty and potential impact of the proposed method, they also identified several areas for improvement.